# Programming DNA Reaction Networks Using Allosteric DNA Hairpins

**DOI:** 10.3390/biom13030481

**Published:** 2023-03-05

**Authors:** Rui Qin, Shuang Cui, Xiaokang Zhang, Peijun Shi, Shihua Zhou, Bin Wang

**Affiliations:** 1Key Laboratory of Advanced Design and Intelligent Computing, Ministry of Education, School of Software Engineering, Dalian University, Dalian 116622, China; 2School of Computer Science and Technology, Dalian University of Technology, Dalian 116024, China

**Keywords:** DNA reaction networks, allosteric DNA hairpins, reversible regulation, functional nucleic acids

## Abstract

The construction of DNA reaction networks with complex functions using various methods has been an important research topic in recent years. Whether the DNA reaction network can perform complex tasks and be recycled directly affects the performance of the reaction network. Therefore, it is very important to design and implement a DNA reaction network capable of multiple tasks and reversible regulation. In this paper, the hairpin allosteric method was used to complete the assembly task of different functional nucleic acids. In addition, information conversion of the network was realized. In this network, multiple hairpins were assembled into nucleic acid structures with different functions to achieve different output information through the cyclic use of trigger strands. A method of single-input dual-output information conversion was proposed. Finally, the network with signal amplification and reversible regulation was constructed. In this study, the reversible regulation of different functional nucleic acids in the same network was realized, which shows the potential of this network in terms of programmability and provides new ideas for constructing complex and multifunctional DNA reaction networks.

## 1. Introduction

Biochemical reaction networks play an important role in living systems [1]. They not only regulate cellular genetics and metabolism but also enable organisms to respond to external stimuli to maintain their stability [2]. Inspired by biochemical reaction networks, rational programming of artificial molecular reaction networks using nanotechnology has attracted increasing attention. Using programmable artificial molecular reaction networks, the reaction paths of complex networks can be effectively designed [3], and the transformation of information can be controlled, which enables these networks to perform their target functions. Programmable artificial molecular reaction networks are widely used in molecular self-assembly [4,5], disease diagnosis and treatment [6,7,8], and neural networks [9,10]. Therefore, it is necessary to design and study programmable artificial molecular reaction networks.

DNA molecules follow the principle of Watson–Crick base complementary pairing [11], which provides strong programmability, high storage [12,13,14], and molecular recognition ability [15,16]. DNA is an important material for constructing artificial molecular reaction networks. DNA-based molecular reaction networks can be constructed with different regulatory strategies, such as toehold mediation [17,18], ion trigger [19,20], light control [21,22], pH control [23,24,25], and enzyme drive [26]. Among them, toehold-mediated DNA strand displacement is an important technique for programming DNA reaction networks, and it can be used to design catalytic [27], inhibitory [28], and dissipative [29] networks. In a toehold-mediated DNA strand displacement reaction network, the bare single-stranded DNA has poor stability and easily self-hybridizes to cause signal crosstalk. As a result, a large number of by-products are formed. However, hairpin structures [30,31] formed by single-stranded DNA fold back onto themselves, which allows complementary base pairs to hybridize to form stable structures and reduces the DNA domains exposed to the solution. Therefore, using hairpin structures to construct DNA reaction networks not only avoids its hybridization on an experimental time scale [32] but also protects the important information in the stem of the hairpin. Only through the hairpin allosteric structure can information be converted and transmitted, thus ensuring the DNA reaction network proceeds in an orderly manner.

In recent years, research on DNA networks based on allosteric hairpin programming has grown. These networks are often based on programmable designs such as catalytic hairpin reactions (CHAs) [33,34,35] and mixed chain reactions (HCRs) [36,37]. These hairpin-based programmable networks show the characteristics of simple design and low background noise [38], which lead to promising application prospects. A CHA’s substrate contains multiple complementary base domains, but the complementary DNA domains are “locked” in the hairpin structure. When no trigger strand exists, the hairpin structure is in a stable state. When a trigger strand is added to the system, multiple hairpins are triggered in turn and then assembled. It is worth noting that the locked DNA domains with special sequences in the hairpin can be constructed into functional nucleic acid structures by CHAs [39]. Among them, aptamer [40], Mg^2+^-ion-dependent DNAzyme [41], and hemin/G-quadruplex DNAzyme [42], as functional nucleic acids combined with CHAs, have been used in logical calculations [43], molecular detection [44,45,46], and biosensing [47,48]. As a result, the combination of functional nucleic acids and CHAs can contribute to building a complex DNA network of allosteric hairpin programming that is suitable for many application scenarios. However, there are challenges in achieving the combination of multiple functional nucleic acids and CHAs in DNA reaction networks.

Here, a strategy of programming a DNA reaction network using allosteric hairpins is proposed. Two functional nucleic acids are integrated into the same network, and reversible conversion of the network is realized using the method of a trigger strand that controls the hairpin allosteric. First, multiple hairpin hybridizations containing DNAzyme subunits are triggered by multiple inputs to generate nucleic acid structures with different functions. Then, a method of single-input dual-output information conversion is proposed. Finally, allosteric hairpins are controlled by inhibition strands and disinhibition strands, which realize the reversible conversion of the network. Due to the use of a hairpin structure for the network, the crosstalk between the DNA strands involved in the reaction is reduced. In addition, with the assistance of a CHA mechanism, two functional nucleic acid structures can be assembled in a hairpin allosteric manner and, according to the demand timely control, network for completing complex tasks. Our network demonstrates potential in terms of programmability and information processing and provides a way to construct complex and versatile DNA reaction networks.

## 2. Materials and Methods

### 2.1. Materials and Chemical Reagents

All DNA primers were purchased from Sangon Biotech. Co., Ltd. (Shanghai, China). Unmodified primers were purified using polyacrylamide gel electrophoresis (PAGE), and primers with fluorophore 6-carboxyfluorescein (6-FAM) and quencher Black Hole Quencher 1 (BHQ1) or RNA base modification were purified using high-performance liquid chromatography (HPLC). All the DNA primers needed for the experiment were dissolved in ultrapure water and quantified using a Nanodrop 2000 spectrophotometer. (Thermo Fisher Scientific Inc., Waltham, MA, USA). In addition, 2,2’-azinobis (3-ethylbenzothiazoline-6-sulfonic acid ammonium salt) (ABTS) was purchased from Tci Development Co., Ltd. (Shanghai, China). Hydrogen peroxide was purchased from Jiangxi Caoshanhu Disinfection Co., Ltd. (Jiangxi, China). Hemin and dimethyl sulfoxide were purchased from Sangon Biotech Co., Ltd. (Shanghai, China). Other chemicals were of reagent grade and used without further purification.

### 2.2. DNA Assembly

All DNA hairpins were annealed using the same annealing procedure. DNA strands were annealed in a 1× TAE/Mg^2+^ buffer (40 mM Tris, 20 mM acetic acid, 100 mM NaCl, 1 mM EDTA∙2Na, and 12.5 mM Mg (OAc) 2; pH 8.0). The solution was heated to 95 °C for 10 min and then cooled to 25 °C at 1 °C per minute and preserved at 25 °C for 1.5 h. The trigger strand was added to the annealed hairpins to realize the assembly of the DNAzyme.

### 2.3. Native PAGE

First, 15 pmol hairpin was added to 30 μL 1× TAE/Mg^2+^ buffer, and then 15 pmol trigger strand was added. After letting the reaction proceed for 4 h, 5 μL of 60% glycerol solution was added, and then the system was analyzed on a 12% native polyacrylamide gel. The 12% native polyacrylamide gel was prepared and run on a BIO-RAD electrophoresis piece of apparatus (BIO-RAD Co., Hercules, CA, USA) at a constant voltage of 80 V in 1× TAE/Mg^2+^ buffer for 2.5 h. After the gel electrophoresis was completed, the gel was stained in Stains-All for 1 h. Then, the gel was faded using natural light irradiation and imaged using a Canon scanner.

### 2.4. Fluorescence Spectroscopy

The fluorescence results were obtained using a TECAN Microplate Reader Spark 20M (Tecan Trading Co., Grödig Australia). The fluorescence intensity was recorded per minute. The fluorescence analysis method for the Mg^2+^-ion-dependent DNAzyme was as follows: First, 15 pmol hairpins H1, H2, and H3, the reporter strand, and different concentrations of trigger strand T1 were added to 40 μL 1× TAE/Mg^2+^ buffer for 4 h, and then the fluorescence of the FAM fluorophore between 500 and 600 nm was recorded at 25 °C. The signal analysis method for the hemin/G-quadruplex DNAzyme was as follows: First, 40 pmol of the hairpins H4, H5, and H6; KCL (4 μL, 500 mM); 0.8 μL hemin (50 μM); and different concentrations of trigger strand T2 were reacted in 40 μL 1× TAE/Mg^2+^ buffer for 4 h. Then, ABTS^2−^ (16 μL, 2.5 mM), H_2_O_2_ (6.4 μL, 50 mM), and ultrapure water (17.6 μL) were added. The final measured volume was 80 μL. The absorbance of ABTS^•−^ between 400 and 500 nm was recorded at 25 °C.

## 3. Results

### 3.1. The Principle and Verification of the Mg^2+^-Ion-Dependent DNAzyme Dynamic Assembly Module

Here, we first propose a DNA reaction network assembled from hairpins into a Mg^2+^-ion-dependent DNAzyme (DNAzyme 1). As shown in Figure 1a, this reaction network is based on three hairpin structures, in which the subunit sequence of the DNAzyme is hidden in the stems of the hairpins H2 and H3, and the other hairpin H1 acts as a bridge connecting the two DNAzyme 1 subunits. When the trigger strand T1 is added, the hairpin H1 is opened through toehold-mediated strand displacement to form the T1-H1 complex, which exposes the toehold sequence connected with H2 and H3. Then, H1 is complementarily paired with the hairpins H2 and H3 through the toehold (ring). Two inactive DNAzyme 1 subunits are assembled into DNAzyme 1, which can recognize and continuously digest the RNA-modified reporter strand D to generate fluorescence signals. In addition, in the process of combining H2 and H1, the trigger strand T1 is displaced, which realizes the regeneration of the trigger strand T1 and acts as the catalyst in the whole DNA reaction network. The DNA reaction network is made up of three hairpins H1, H2, and H3 and a DNA reporter strand D containing ribonucleic acid. The reporter strand D was labeled with fluorescein/quencher (F/Q) at its 5′ and 3′ ends (F: 6′-FAM; Q: BHQ1), which achieves luminescence quenching of the fluorophore through an effective fluorescence resonance energy transfer process. The reaction is illustrated in the abstract diagram in Figure 1b, where the brown-filled circles represent the three hairpins, and the green circle represents the trigger strand T1 as the catalyst that catalyzes the formation of DNAzyme 1. The blue-filled circles represent the formation of DNAzyme 1, and the formed DNAzyme 1 continues to digest the reporter strand.

The feasibility of the DNAzyme1 dynamic assembly module was investigated using native PAGE. As shown in Figure 1c, in the absence of a corresponding trigger (lane 5), there is no crosstalk between the reactants, which can be observed by the fact that only three separate bands corresponding to H1 (lane 2), H2 (lane 3), and H3 (lane 4) are present. When the trigger strand was added to the hairpin mixture (lane 6), a new band with lower electrophoretic mobility was observed, which does not correspond to T1 (lane 1). The new band was proven to be the generated DNAzyme 1 by the auxiliary verification of Figure 1d. As shown in Figure 1d, the fluorescence signal increased significantly after the addition of the catalyst T1. To determine the catalytic effect of the trigger strand T1, a series of different concentrations of the trigger strand T1 were used for control experiments. As shown in Figure 1d, a T1 concentration of 0.3 μM (0.8×) had the same effect as that of 0.375 μM (1×). The trigger strand T1 signal amplification intensity was analyzed in Appendix A. The effect of T1 as a catalyst is further illustrated in Appendix A.

### 3.2. The Principle and Verification of the Hemin/G-quadruplex DNAzyme Dynamic Assembly Module

Here, we propose a DNA reaction network assembled from hairpins into the hemin/G-quadruplex DNAzyme (DNAzyme 2). As shown in Figure 2a, this reaction network is based on three hairpin structures, in which the subunit sequence of the DNAzyme is hidden in the stems of the hairpins H5 and H6, and the other hairpin H4 acts as a bridge connecting the two DNAzyme 2 subunits. When the trigger strand T2 is added, the hairpin H4 is opened through toehold-mediated strand displacement to form the T2-H4 complex, which exposes the toehold sequence connected with H5 and H6. Then, H4 is complementarily paired with the hairpins H5 and H6 through the toehold (ring). Two inactive DNAzyme subunits are assembled into DNAzyme 2, which can catalyze the oxidation of ABTS^2-^ to the blue-green ABTS^•−^ product under the action of H_2_O_2_, thus generating the colorimetric readout signal. In addition, in the process of combining H5 and H4, the trigger strand T2 is displaced, which realizes the regeneration of the trigger strand T2 and acts as the catalyst in the whole DNA reaction network. The reaction is illustrated in the abstract diagram in Figure 2b, where the brown-filled circles represent the three hairpins, and the green circle represents the trigger strand T2 as the catalyst that catalyzes the formation of DNAzyme 2. The blue-filled circles represent the formation of DNAzyme 2, and the formed DNAzyme 2 catalyzes the oxidation of ABTS^2-^ to the blue-green ABTS^•−^ product under the action of H_2_O_2_.

The feasibility of the DNAzyme dynamic assembly module was investigated using native PAGE. As shown in Figure 2c, in the absence of a corresponding trigger (lane 6), there is no crosstalk between the reactants, which can be observed by the fact that three separate bands corresponding to H4 (lane 2), H5 (lane 3), and H6 (lane 4) are observed, and no new bands are observed. When the trigger strand was added to the hairpin mixture (lane 7), a new band with lower electrophoretic mobility was observed, which does not correspond to T2 (lane 1) or H4-H5 (lane 5). The new band was proven to correspond to the generated DNAzyme 2 using the auxiliary verification of Figure 2d. As shown in Figure 2d, a colorimetric readout signal was generated after the addition of the catalyst T2. To determine the catalytic effect of the trigger strand T2, a series of different concentrations of the trigger strand T2 were used for control experiments. As shown in Figure 2d, a T2 concentration of 0.8 μM (0.8×) and 0.6 μM (0.6×) had the same effect as that of 1 μM (1×). We explore the conditions for the dynamic assembly of G4 in Appendix A and Appendix A. The trigger strand T2 signal amplification intensity was analyzed in Appendix A. The effect of T2 as a catalyst is further illustrated in Appendix A.

### 3.3. The Principle and Verification of the Single-Input Dual-Output Module

Here, we implement single-input dual-output signal conversion in the DNA reaction network. Through the input of a single strand of DNA, the dynamic assembly of the two DNAzymes is completed, which then realizes the output of two signals, as shown in Figure 3a. The reaction network is based on six hairpin structures, and the two inactive DNAzyme subunits are assembled into DNAzymes 1 and 2 under the guidance of the trigger strand S. DNAzyme 1 can recognize and continuously digest the RNA-modified reporter strand D to generate fluorescence signals. DNAzyme 2 can catalyze the oxidation of ABTS^2-^ to the blue-green ABTS^•−^ product under the action of H_2_O_2_, thus generating the colorimetric readout signal. The reaction is illustrated in the abstract diagram in Figure 3b, where the brown-filled circles represent the six hairpins, and the green circle represents the trigger strand S catalyzing the formation of DNAzymes 1 and 2. The blue-filled circles represent the formation of DNAzymes 1 and 2, and the formed DNAzyme 1 can recognize and continuously digest the RNA-modified reporter strand D to generate fluorescence signals. DNAzyme 2 catalyzes the oxidation of ABTS^2−^ to the blue-green ABTS^•−^ product under the action of H_2_O_2_. Figure 3c is an overview diagram of the single-input dual-output signal conversion. Using the strand S as the input, the two signals of DNAzymes 1 and 2 were obtained after the action of the six hairpins.

The verification of the single-input dual-output module can be demonstrated as follows: As shown in Figure 3d, DNAzymes 1 and 2 show a lower fluorescence or absorbance reading in the absence of the input. To prove single-input dual-output signal conversion can be achieved, we need to prove that the single-input single-output signal conversion can be achieved separately (Appendix A). When the input was added, the fluorescence or absorbance rose. This means that the proposed method of the single-input dual-output information conversion was verified.

### 3.4. The Principle and Verification of the Reversible Regulatory DNA Reaction Network

#### 3.4.1. The Principle of the Reversible Regulatory DNA Reaction Network

Reversibly regulated DNA reaction networks are programmed using allosteric hairpins. The successful design of the above three modules guided the reaction network constructed by multiple hairpins containing two different DNAzymes, and the reversible regulation operation of the two DNAzymes was realized, as shown in Figure 4. First, the DNA reaction network was guided by a trigger strand to synthesize DNAzymes 1 and 2, which is case 1 in Figure 4a. In case 1, the hairpins H1, H2, H3, H4, H5, and H6 and the fluorescence reporter strand D were substrates. The DNAzyme subunit sequence was concealed. When the trigger strand S was the input, multiple hairpins were guided to assemble DNAzyme 1 (H1-H2-H3) and DNAzyme 2 (H4-H5-H6) simultaneously. Then, DNAzyme 1 continuously digested the fluorescent substrates, resulting in fluorescence changes. DNAzyme 2 catalyzed the oxidation of ABTS^2-^ to the blue-green ABTS^•−^ product under the action of H_2_O_2_, as shown in Figure 3d.

The regulatory module of the DNA reaction network was realized by inhibiting and disinhibiting the two types of DNAzymes. To enable the suppression operation of the DNA reaction network, the hairpins H1 and H4 were pre-designed as two toeholds that can hybridize with the inhibitors C1 and C2, respectively. When the inhibitor C1 was added to case 1, the DNA reaction network transformed into the module in case 2. Inhibitor C1 through six nt toeholds at the 3′ end of H1 displaced H2. Due to the self-tension of the hairpin, the displaced H2 strand bent into a stable hairpin shape. In other words, the hairpin H2 in the DNAzyme 1 was already formed in the DNA reaction network and was replaced to form the H1-H3-C1 complex, resulting in the inactivation of DNAzyme 1. In addition, after the addition of the inhibitor C1, complex C1-H1-H3 is formed instead of DNAzyme 1 because the structures that did not form DNAzyme 1 preferentially reacted with C1. In this case, the strand S could still trigger the formation of DNAzyme 2 as before, but the inhibitor C1 inhibited the formation of DNAzyme 1. Conversely, in order to achieve the disinhibition operation of the DNA reaction network and make DNAzyme 1 re-enter the working state, the inhibitor C1 was pre-designed as a toehold that can hybridize with the de-inhibitor C1*. (All the asterisks in the figure indicates the complementary strand of the corresponding strand.) By adding the de-inhibitor C1*, the DNA reaction network returned to the working state from the previous DNAzyme 1-inhibited state. At this time, the inhibitor C1 in the H1-H3-C1 complex underwent toehold-mediated strand replacement with the de-inhibitor C1*, and C1 broke away from the complex to form the C1-C1* complex. At the same time, the H1-H3 complex opened the hairpin H2 again. DNAzyme 1 was re-constituted, and the active DNAzyme 1 again digested the fluorescence reporter strand D. In this process, the module of DNAzyme 2 was activated normally, and the activity of the DNAzyme 1 module was regulated with the addition of the inhibition strands.

Similarly, when the inhibitor C2 was added to case 1, the DNA reaction network transformed into the module in case 3. The inhibitor C2 through six nt toeholds at the 3′ end of H4 displaced H5. Due to the self-tension of the hairpin, the displaced H5 strand bent into a stable hairpin shape. In other words, the hairpin H5 in the DNAzyme 2 was already formed in the DNA reaction network and was replaced to form the H4-H6-C2 complex, resulting in the inactivation of DNAzyme 2. In addition, after the addition of the inhibitor C2, complex C2-H4-H6 was formed instead of DNAzyme 2 because the structures that did not form DNAzyme 2 preferentially reacted with C2. In this case, the strand S could still trigger the formation of DNAzyme 1 as before, but the inhibitor C2 inhibited the formation of DNAzyme 2. Conversely, in order to achieve the disinhibition operation of the DNA reaction network and make DNAzyme 2 re-enter the working state, the inhibitor C2 was pre-designed as a toehold that can hybridize with the de-inhibitor C2*. By adding the de-inhibitor C2*, the DNA reaction network returned to the working state from the previous DNAzyme 2-inhibited state. At this time, the inhibitor C2 in the H4-H6-C2 complex underwent toehold-mediated strand replacement with the de-inhibitor C2*, and C2 broke away from the complex to form the C2-C2* complex. At the same time, the H4-H6 complex opened the hairpin H5 again. The DNAzyme 2 was re-constituted, and the active DNAzyme 2 was formed. In this process, the module of DNAzyme 1 was activated normally, and the activity of the DNAzyme 2 module was regulated with the addition of the inhibition strands.

The reaction is illustrated in the abstract diagram in Figure 4b, which represents the transformation of the structure in case 1 during the regulation process. The brown-filled circles represent the six hairpins, and the green circle represents the trigger strand S as the catalyst that catalyzed the formation of the two DNAzymes. The blue-filled circles represent the formation of the two DNAzymes. When C1 was added to case 1, the module of DNAzyme 2 was activated normally, and the activity of the DNAzyme 1 module was regulated with the addition of C1 and C1*, leading to the recycling of H2. When C2 was added to case 1, the module of DNAzyme 1 was activated normally, and the activity of the DNAzyme 2 module was regulated with the addition of C2 and C2*, leading to the recycling of H5.

#### 3.4.2. Verification of the Reversible Regulatory DNA Reaction Network

Verification of the reversible regulation of the programmed DNA reaction network was conducted in three stages as follows: first, multiple reversible regulations of DNAzyme1 in a network; second, multiple reversible regulations of DNAzyme2 in a network; and finally, mixed regulation of DNAzyme1 and DNAzyme2 in a network. If we want to achieve hybrid regulation in the same network, then we need to ensure that reversible regulation can be achieved in a single network (Appendix A). Figure 5a shows the fluorescence changes produced by the addition of the inhibitor C1 and de-inhibitor C1* at different time points (DNAzyme 1 digests the reporter strand D). We put all of the substrate (the hairpins H1, H2, H3, H4, H5, and H6 and the fluorescence reporter strand D) in the solution, and first, we added the trigger strand S; second, after an hour and a half, we added the inhibitor C1; third, after another hour and a half, we added the de-inhibitor C1*; and we performed that three times. The trigger strand S was added to the reaction for 1.5 h, and we can see that the fluorescence increased significantly within the 1.5 h. When C1 was added at 1.5 h, DNAzyme 1 activity was lost. The hairpin H2 with a part of the DNAzyme 1 subunit structure was replaced, and the increase in fluorescence stopped. It can be seen from Figure 5a that the fluorescence still increased at a very slow rate. The reason why the fluorescence did not stop increasing immediately is that it took time for the inhibitor C1 to completely displace H2. When the inhibitor C1 was not completely replaced, there was still a part of DNAzyme 1 that digested the reporter strand D, which produced fluorescence. When the de-inhibitor C1* was added at 3 h, the fluorescence increased again at a high rate, and the same operation was performed in three cycles.

DNAzyme 2 catalyzed the oxidation of ABTS^2-^ to the blue-green ABTS^•−^ product under the action of H_2_O_2_, as shown in Figure 5b. The absorbance changes in ABTS^•−^ at different time points with the addition of the inhibitor C2 and de-inhibitor C2* are also shown. The absorbance increased significantly within 1.5 h after the triggering agent was added. When C2 was added at 1.5 h, the DNAzyme 2 was inactivated, and the hairpin H5 with a part of the DNAzyme 2 subunit structure was replaced, and the absorbance began to decline. When C2* of the disinhibition strand was added at 3 h, the absorbance of the ABTS^•−^ began to increase again, and the same operation was performed in three cycles. In Figure 5a,b, the inhibitory effect and the disinhibitory effect were both as expected.

The mixed regulation of this network is shown in Figure 5c,d. The trigger strand S was added to the reaction for 1.5 h, and the fluorescence and absorbance increased significantly within 1.5 h. When C2 was added at 1.5 h, DNAzyme 2 activity was lost. The hairpin H5 with a part of the DNAzyme 2 subunit structure was replaced, and the increase in absorbance stopped. When the de-inhibitor C2* was added at 3 h, the absorbance increased again at a high rate. However, the fluorescence intensity of DNAzyme 1 was not affected at all during this process. Then, we regulated the activity of DNAzyme 1; when the inhibitor C1 was added at 4.5 h, DNAzyme 1 activity was lost. The hairpin H2 with a part of the DNAzyme 1 subunit structure was replaced, and the increase in fluorescence stopped. When the de-inhibitor C2* was added at 6 h, the fluorescence increased again at a high rate. However, the absorbance intensity of DNAzyme 2 was not affected at all during this process.

In addition, before the reaction, that is, before the trigger strand S was added, if C1 or C2 was added to the substrate, the inhibitor C1 showed no effect on DNAzyme 2, and the inhibitor C2 showed no effect on DNAzyme 1. We performed a controlled experiment, as shown in Figure 6. ABCD indicates that the absorbance and fluorescence values, respectively, were measured after reacting in the shown substrate for 4 h. A means that only the substrate hairpins H1, H2, H3, H4, H5, and H6 and the fluorescence reporter strand D were added to the solution. B represents the addition of the trigger strand S to the substrates. C and D represent the addition of different inhibition strands on the basis of B. It can be seen from C and D that when the inhibitor C1 was added, a lower fluorescence value could be read, but the absorbance was the same as that of group B without the inhibitor. When the inhibitor C2 was added, a lower absorbance could be read, but the fluorescence value was the same as that of group B without the inhibitor. We can see that when DNAzyme 1 was suppressed, C1 showed no effect on DNAzyme 2. DNAzyme 2 was suppressed, and C2 had no effect on DNAzyme 1. This indicates that the above conclusions are valid.

## 4. Discussion

As DNA reaction networks are required to perform complex tasks and be recycled, the functional diversity, regulative capability, and reaction rate of the network together determine the performance of the network. This study designed and implemented a DNA reaction network by using allosteric hairpins and then discussed the performance of the network from the above three aspects.

It introduced functional nucleic acids to enrich the functional diversity of DNA reaction networks. Previous researchers have developed DNA reaction networks that could perform specific functions by introducing functional nucleic acids, i.e., biosensors for molecular detection [49] and DNAzyme networks for logical operations [50]. This study integrated two functional nucleic acids into the same network. The increase in the numbers of functional nucleic acids could not only extend the signal output manners but also enhance the function of the network. Gao et al. [51] proposed a DNAzyme-based biosensor network, which uses a strategy that four substrate hairpins to generate a kind of DNAzyme structure. Following this way, we used three substrate hairpins in our generation strategy. Fewer substrate hairpins can reduce the total number of oligonucleotides in the DNA reaction network and decrease crosstalk between the DNA hairpins. Moreover, it realized the generation of two functional nucleic acid structures based on the strategy, which proved that the extensible strategy would facilitate the construction of complex multifunctional DNA reaction networks.

To improve the regulative capability of the network, it realized the reusable DNA network by designing reversible regulation of the inhibitor and de-inhibitor. As Harding et al. [52] said, iteration reduces the amount of physical codes needed for computers, and reusability will reduce the total number of oligonucleotides needed for DNA networks. Therefore, the reversible regulatory ability is important for designing networks. This study implemented multiple reversible regulations in the network (Figure 5a,b). The reversible regulation function does not require additional substrates in the regulation process for a material cost saving. In addition, the functional nucleic acid sequences do not interfere with each other during the regulatory process due to a reasonable reaction environment and DNA sequence design (Figure 5c, d). When it reversibly regulated DNAzyme 1, DNAzyme 2 would not be disturbed. Similarly, DNAzyme 2 would not be disturbed by DNAzyme 1.

To accelerate the reaction rate, a CHA mechanism was used to catalyze the DNA reaction to increase the reaction rate and amplify the signal. Compared with this network without a CHA mechanism, higher fluorescence values were obtained at the same reaction time as with the CHA mechanism (Appendix A, green bar). The trigger strand acted as a catalyst for the DNA reaction network while effectively guiding these substrate hairpins for integration, then catalytically assembled a large number of DNAzymes while increasing the reaction rate.

## 5. Conclusions

This study used allosteric hairpins to design and implement a DNA reaction network that is capable of multiple tasks and reversible regulation. Specifically, it constructed a DNA reaction network with a CHA reaction, which can generate and disassemble two functional nucleic acids. The experimental results show that the DNA reaction network possesses information processing ability and programmability. In addition, by hiding different functional nucleic acid sequences, such as i-motif, aptamer, and DNAzyme in hairpins, it can further increase the scale of the reaction network and provide a new idea for the construction of complex multifunctional DNA reaction networks. However, as the scale of the DNA reaction network expands, it may increase uncertainty in the reaction network, i.e., cross-interference between signals caused by leakage reactions. Therefore, we will further explore the influence of scale expansion on the reaction network accordingly.

## Figures and Tables

**Figure 1 biomolecules-13-00481-f001:**
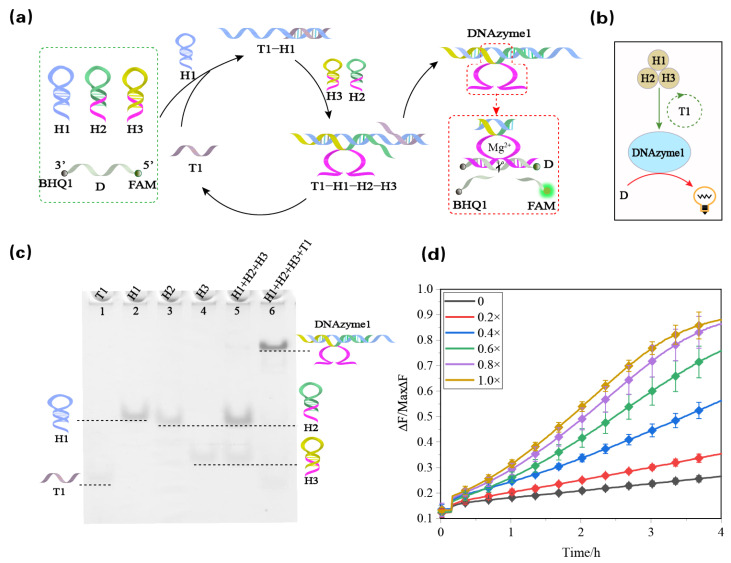
(**a**) Schematic diagram of the Mg^2+^-ion-dependent DNAzyme (DNAzyme 1) assembled from hairpins. (**b**) Abstract diagram of DNAzyme 1 assembled from hairpins. The brown circle represents the three hairpins; the green circle without color filling represents the triggering strand T1 as the catalyst to guide the formation of DNAzyme 1; and the blue circle represents the formed DNAzyme 1 that continuously digests the reporter strand. (**c**) DNAzyme 1 catalytic reaction module was analyzed using polyacrylamide gel with 12% PAGE. The hairpin and single strand involved are marked above the lane. Lane 1: trigger strand T1; lane 2: hairpin H1; lane 3: hairpin H2; lane 4: hairpin H3; lane 5: hairpins H1, H2, and H3; lane 6: trigger strand is added to the three hairpins. (**d**) Fluorescence intensity with time when different concentrations of catalyst T1 were input.

**Figure 2 biomolecules-13-00481-f002:**
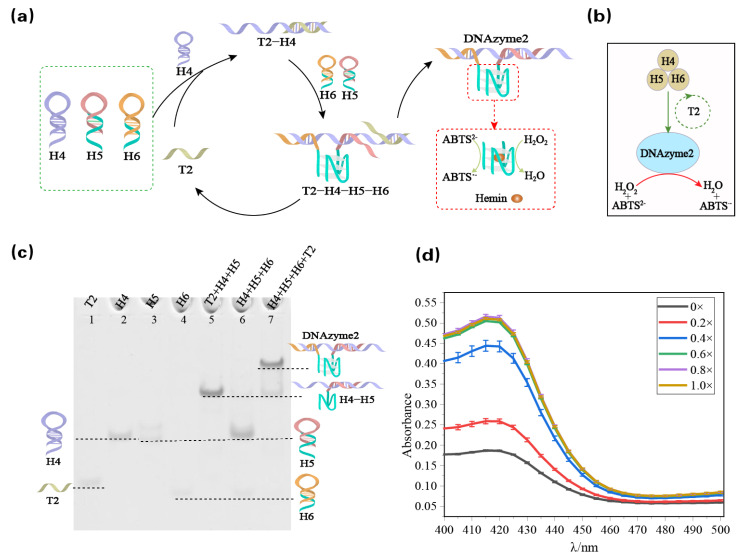
(**a**) Schematic diagram of the hemin/G-quadruplex DNAzyme (DNAzyme 2) assembled from hairpins. (**b**) Abstract diagram of DNAzyme 2 assembled from hairpins. The brown circle represents the three hairpins; the green circle without color filling represents the triggering strand T2 as the catalyst to guide the formation of DNAzyme 2; and the blue circles represent the formation of DNAzyme 2 that catalyzes the transformation of ABTS^2−^ to ABTS^•−^. (**c**) DNAzyme 2 catalytic reaction module was analyzed using polyacrylamide gel with 12% PAGE. The hairpin and single strand involved are marked above the lane. Lane 1: trigger strand T2; lane 2: hairpin H4; lane 3: hairpin H5; lane 4: hairpin H6; lane 5: complex of H4 and H5; lane 6: hairpins H4, H5, and H6; lane 7: trigger strand is added to the three hairpins. (**d**) Absorbance image of the resulting ABTS^•−^ when different concentrations of catalyst T2 are input.

**Figure 3 biomolecules-13-00481-f003:**
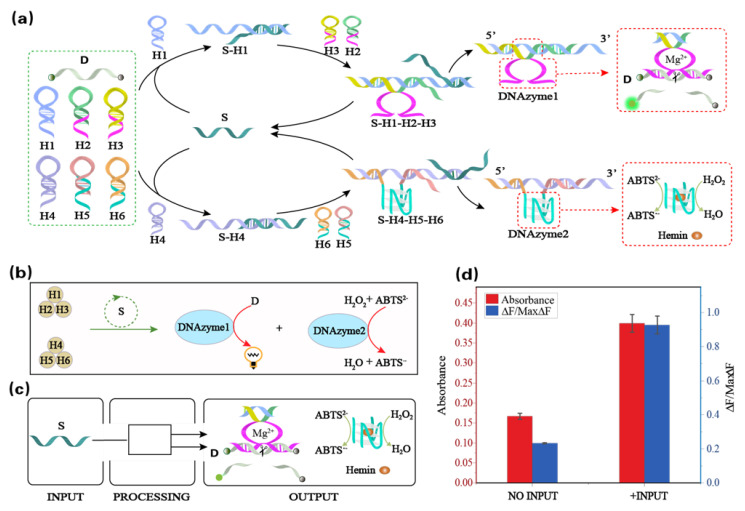
(**a**) Schematic diagram of the single-input dual-output signal conversion. (**b**) Abstract diagram of the single-input dual-output signal conversion. (**c**) Overview diagram of the single-input dual-output signal conversion. (**d**) Bar graph with two output signals with or without the input. The fluorescence (at λ = 520 nm) intensity of DNAzyme 1 and the absorbance of ABTS^•−^ (at λ = 420 nm) by DNAzyme 2 are shown.

**Figure 4 biomolecules-13-00481-f004:**
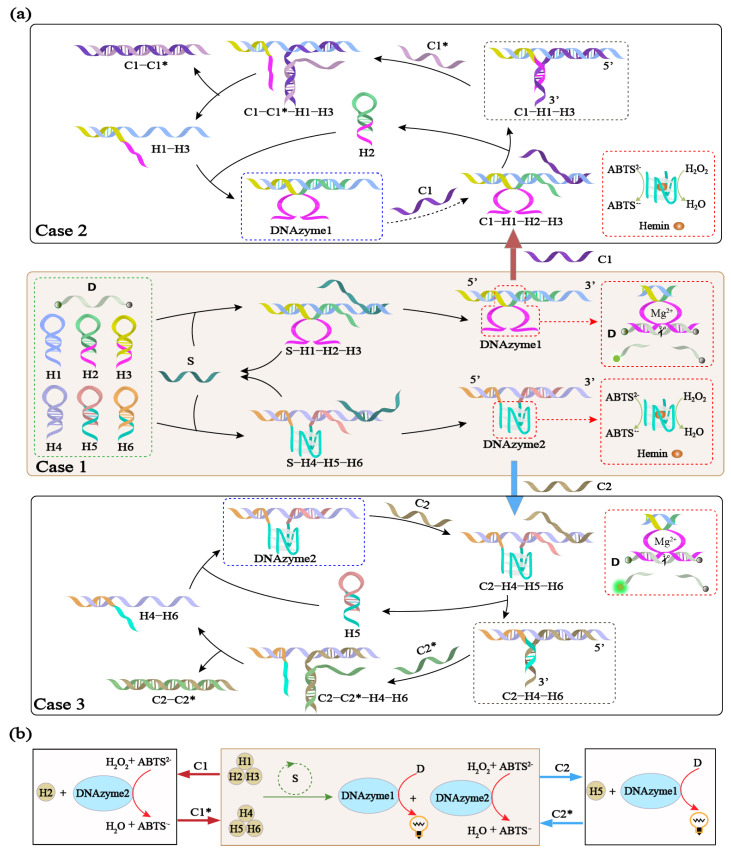
(**a**) Schematic diagram of the reversible regulation of DNAzymes 1 and 2 using allosteric hairpins. Case 1 presents the non-reactive module consisting of the hairpins H1, H2, H3, H4, H5, and H6 and the fluorescence reporter strand D. Hemins are also present. Subjecting case 1 to the triggers S leads to the dynamic formation of DNAzymes 1 and 2. Treatment of case 1 with the inhibitor C1 substitutes the hairpin H2, leading to the reaction module in case 2. Then, only DNAzyme 2 is generated in case 2, while the formation of DNAzyme 1 is blocked. Treatment of case 2 with the inhibitor C1*, leads to the reaction module in case 1. Treatment of case 1 with the inhibitor C2 substitutes the hairpin H5, leading to the reaction module in case 3. Then, only DNAzyme1 is generated in case 2, while the formation of DNAzyme 2 is blocked. Treatment of case 3 with the inhibitor C2*, leads to the reaction module in case 1. (**b**) Abstract diagram of the reversible regulation of DNAzymes 1 and 2 using allosteric hairpins.

**Figure 5 biomolecules-13-00481-f005:**
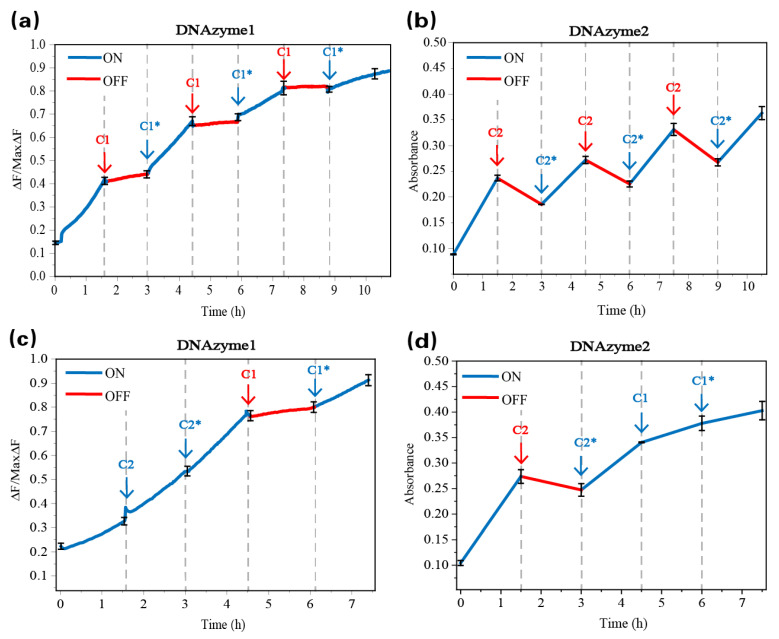
Time-dependent fluorescence and absorbance changes of DNAzymes 1 and 2. (**a**) Reversible conversion between cases 1 and 2 is controlled by the inhibitor C1 and de-inhibitor C1* and the inhibition and activation of DNAzyme 1 activity. (**b**) Reversible conversion between cases 1 and 3 is controlled by the inhibitor C2 and de-inhibitor C2* and the inhibition and activation of DNAzyme 2 activity. (**c**) The reversible conversion between cases 1 and 3 and between cases 1 and 2 is operated simultaneously within a specified time. The signal changes of DNAzyme 1 and (**d**) DNAzyme 2 were observed.

**Figure 6 biomolecules-13-00481-f006:**
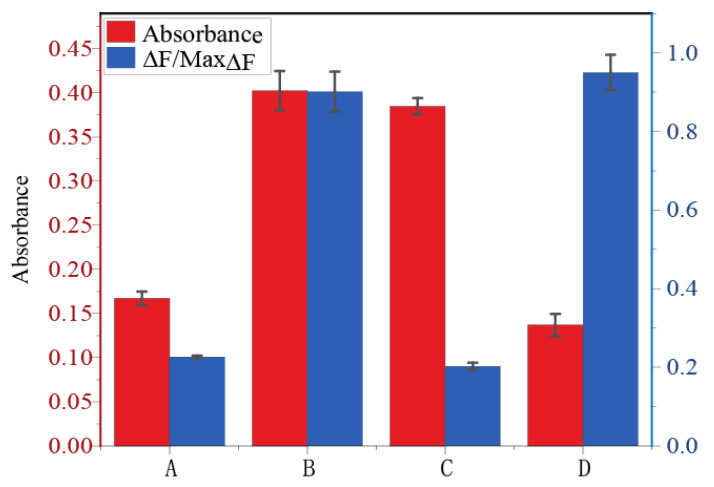
Changes in fluorescence intensity of DNAzyme 1 and the absorbance of ABTS^•−^ by DNAzyme 2 under different conditions. A: substrate; B: substrate + S; C: substrate + S + C1; D: substrate + S + C2. The left Y-axis of the figure represents the absorbance of G4, and the right Y-axis represents the fluorescence value.

## Data Availability

Not applicable.

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
