# Peer review of "Programming DNA Reaction Networks Using Allosteric DNA Hairpins"

_biomolecules, 2023, doi:10.3390/biom13030481_

Round 1
Reviewer 1 Report
In this study, a strategy for programming DNA reaction networks is proposed, aiming to realize complex functions and reversible regulations. Based on the strategy, DNA chemical reaction networks are constructed by catalytically assembling multiple kinds of DNA hairpins, in which the signals can be amplified and dynamically regulated. The results of the experiments are clear and convictive. However, there are still some issues should be well explained to meet the publication standard of Biomolecules. Before it is to be published, it is essential to revise the manuscript very carefully.
(1) Section 2.3: The PAGE method is not described clearly. How was the DNA visualized? Were the gels stained? What kind of dyes are used? Explain explicitly to ensure the repeatability of the experiments.
(2) As the authors claimed “The fluorescence analysis method for the Mg2+-ion-dependent DNAzyme” and “The absorbance of ABTS•− between 400 and 500 nm was recorded”. Why are both real-time PCR and optical absorption used? What are they used for?
(3) The G-quadruplex can be catalytically assembled in the Section 3.2 and can be dynamically regulated in the Section 3.4. What is the significance of such a structure that can be controlled to generate and disassembled?
(4) Line 281 and Line 301: The descriptions are confused. “In addition, the remaining hairpins that did not form DNAzyme 1 also did not form DNAzyme 1”, “In addition, the remaining hairpins that did not form DNAzyme 2 also did not form DNAzyme 2”. So, what DNAzymes are actually formed and what are not? The authors must address these confusing statements/facts.
(5) Two H2 hairpins are drawn in the figure 4(a). Do these two H2 hairpins represent the same hairpin, which means that, they are consumed after being generated? Or was one of the hairpins added later? To correct the picture to eliminate this misunderstanding.
(6) In Discussion part, what kinds of “other functional nucleic acid sequences” can be concealed? By concealing these nucleic acid sequences, what functions can be realized and what programmable reaction networks can be built? Suggest some possible examples.
Author Response
Thank you very much for the comments. Please refer to the attachment for detailed replies

Reviewer 2 Report
DNA reaction networks have been developed for use in a variety of logic computing, molecular detection, disease treatment and biosensing scenarios. In this paper, the authors achieved the reversible conversion of two functional nucleic acids in reaction networks by controlling allosteric hairpins. This study provides new strategies for building programmable artificial molecular reaction networks that provide insights into complex logical computations. However, I think the authors are not very accurate in some aspects of their presentation, and my detailed questions and comments are as follows.
Concern #1: Why the hairpins in the substrate do not continue to form DNAzyme 1 or DNAzyme 2 after adding the inhibitor strand (C1 or C2)?
Concern #2: How Mg2+-ion-dependent DNAzyme and hemin/G-quadruplex can coexist and not affect each other in the same DNA reaction network?
Concern #3: (Section 2.3) The PAGE method is not very clear. How was the DNA visualized? How were the gels stained?
Concern #4: Please include the full name of the fluorophore (6-FAM) and the quencher (BHQ1) in the manuscript.
Concern #5: How to design the sequence, or what method was used to design the sequence?
Author Response
Response to Reviewer 2 Comments
Concern #1: Why the hairpins in the substrate do not continue to form DNAzyme 1 or DNAzyme 2 after adding the inhibitor strand (C1 or C2)?
Response 1: Thank you very much for your question. The length of the complementary domain between H1 and C1 is 30 bp, which is 10 bp longer than the complementary domain length of H1 and S. After the addition of inhibitor C1, H1 will preferentially bind to C1 to form stable complex C1-H1-H3, rather than bind to trigger strand S to form DNAzyme 1. In addition, for the DNAzyme 1 formed in the reaction network, C1 displaces hairpin H2 through the toehold domain exposed by DNAzyme 1 to form a stable complex C1-H1-H3. Similarly, when inhibitor C2 is added, complex C2-H4-H6 is formed instead of DNAzyme 2.
Concern #2: How Mg2+-ion-dependent DNAzyme and hemin/G-quadruplex can coexist and not affect each other in the same DNA reaction network?
Response 2: We used suitable reaction conditions to ensure Mg2+-ion-dependent DNAzyme and hemin/G-quadruplex can coexist and not affect each other. The reaction buffer was 1×TAE/Mg2+ containing Hemin and KCL. 1×TAE/Mg2+ buffer is the reaction condition for Mg2+-ion-dependent DNAzyme. This reaction condition is also applicable to hemin/G-quadruplex. Hemin and KCL were added to ensure hemin/G-quadruplex reaction condition, and these two substances had no effect on Mg2+-dependent DNAzyme.
Concern #3: (Section 2.3) The PAGE method is not very clear. How was the DNA visualized? How were the gels stained?
Response 3: Thank you very much for pointing this out. We are very sorry for our negligence of PAGE method description. After the gel electrophoresis was completed, the gel was placed in Stains-All for DNA staining. Then, the gel was faded using natural light irradiation and imaged using a Canon scanner.
We have modified the manuscript in the section 2.3 Line 115-117. The modifications are following:
“……buffer for 2.5 h. After the gel electrophoresis was completed, the gel was stained in Stains-All for 1 h. Then, the gel was faded using natural light irradiation and imaged using a Canon scanner.”
Concern #4: Please include the full name of the fluorophore (6-FAM) and the quencher (BHQ1) in the manuscript.
Response 4: We are very sorry for our negligence of the full name of the fluorophore (6-FAM) and the quencher (BHQ1).
We have modified the manuscript in the section 2.1 Line 91-94. The modifications are following:
“Unmodified primers were purified by polyacrylamide gel electrophoresis (PAGE), and primers with fluorophore 6-Carboxyfluorescein (6-FAM) and quencher Black Hole Quencher 1 (BHQ1) or RNA base modification were purified by high-performance liquid chromatography (HPLC).”
Concern #5: How to design the sequence, or what method was used to design the sequence?
Response 5: All sequences required for the experiment are designed using the NUPACK. NUPACK is a software for the analysis and design of nucleic acid structures. We use it to analyze free energy, simulate the binding of DNA strands under specific conditions, check whether there are other structures or base mismatches, and optimize design sequence.
We have modified the manuscript accordingly in the section 2.1 of Supporting Information:
“……of the DNAzyme. All DNA sequences required for the experiments were designed using the NUPACK. We use it to analyze free energy, simulate DNA strands binding, check base mismatches, and optimize the design sequence.”
Reviewer 3 Report
The article by Rui Qin et al. deals with important issues about DNA networks. In particular, the authors propose a strategy of programming a DNA reaction network using allosteric hairpins. Their construction strategy using the allosteric hairpin is well explained and the experiments used are good for devising a technique of constructing and implementing programming a DNA reaction network capable of multiple tasks and furthermore it can help to develop various reaction networks of the DNA. In conclusion, I think that the paper can be considered for publication without revision.
Best Regards
Author Response
Response to Reviewer 3 Comments
The article by Rui Qin et al. deals with important issues about DNA networks. In particular, the authors propose a strategy of programming a DNA reaction network using allosteric hairpins. Their construction strategy using the allosteric hairpin is well explained and the experiments used are good for devising a technique of constructing and implementing programming a DNA reaction network capable of multiple tasks and furthermore it can help to develop various reaction networks of the DNA. In conclusion, I think that the paper can be considered for publication without revision.
Thank you very much for your reading and approval of our manuscript.
Reviewer 4 Report
The manuscript is interesting, it is about a strategy of programming a DNA reaction network using allosteric hairpins. I have some comments for the authors.
1. I think that the discussion needs to be improved, the results should be compared with previous studies. When i read the discussion it seemed to me a summary of the results.
2. Please add a conclusion section of the most important of your study.
3. I don't see any standard deviations in the graphs. How many replicates did you make of each experiment? if it was a replica, is it enough?
4. some mistakes:
line 95: sometimes (Waltham, MA, USA) you include city, abbreviated state and country, other 113 (California, USA) state and country. Please homologate in the whole manuscript.
line 144: ribonucleic acid (rA). I did not find that rA is used again in the manuscript, it could erase.
Author Response
Thank you very much for the comments. Please refer to the attachment for detailed replies.

Round 2
Reviewer 4 Report
The paper is accepted in present form.